# The Role of Iron-Chelating Therapy in Improving Neurological Outcome in Patients with Intracerebral Hemorrhage: Evidence-Based Case Report

**DOI:** 10.3390/medicina59030453

**Published:** 2023-02-24

**Authors:** Mochamad Iskandarsyah Agung Ramadhan, Shierly Novitawati Sitanaya, Ariadri Hafian Wulandaru Hakim, Yetty Ramli

**Affiliations:** 1Dr. Cipto Mangunkusumo Hospital, Jakarta 10430, Indonesia; 2Department of Neurology, Faculty of Medicine, Universitas Indonesia, Jakarta 10430, Indonesia

**Keywords:** deferoxamine, intracerebral hemorrhage (ICH), iron-chelating therapy, modified Rankin Scale (mRS), National Institutes of Health Stroke Scale (NIHSS)

## Abstract

Current primary intracerebral hemorrhage (ICH) treatments focus on limiting hematoma volume by lowering blood pressure, reversing anticoagulation, or hematoma evacuation. Nevertheless, there is no effective strategy to protect the brain from secondary injury due to ICH. Excess heme and iron as by-products of lysing clots in ICH might contribute to this secondary injury by triggering perihematomal edema. We present a clinical situation of an ICH case where iron-chelating therapy might be beneficial, as supported by scientific evidence. We looked through four databases (Pubmed, Cochrane, Embase, and Google Scholar) to find studies assessing the efficacy of iron-chelating therapy in ICH patients. Validity, importance, and applicability (VIA) of the included articles were appraised using worksheets from the Oxford Centre for Evidence-Based Medicine. Two out of five eligible studies were valid, important, and applicable to our patient. Both studies showed the positive effects of iron-chelating therapy on neurological outcome, as measured by National Institutes of Health Stroke Scale (NIHSS) score and modified Rankin Score (mRS). The beneficial effects of deferoxamine were demonstrated within the moderate volume (10–30 mL) subgroup, with a positive relative risk reduction (RRR) and low number needed to treat (six persons). Based on our appraisal, we considered iron-chelating therapy as an additional therapy for ICH patients, given its benefits and adverse effects. More specific studies using a larger sample size, focusing on moderate-volume ICH, and using standardized neurological outcomes are encouraged.

## 1. Introduction

Over 28% of stroke cases yearly are caused by spontaneous intracerebral hemorrhage (ICH). Fifty-four percent of the cases are suffered by men [1]. Data from the Indonesian Stroke Registry [2] reported 1603 (29%) ICH cases from 2012 to 2014, with male and elderly (>60 years old) predominance of 56.52% and 36.77%, respectively. ICH is a devastating cerebral event for patients and their families because patients usually present with worse conditions (e.g., worse Glasgow Coma Scale score) and slightly higher mortality than in ischemic stroke [3]. Even after successfully enduring critical states, survivors may suffer from lifelong disabilities. Disability-adjusted life years (DALYs) were 832.77 per 100,000 cases per year, and almost 69 million years of healthy state were lost [1]. A successful ICH treatment for saving or protecting perilous tissues from secondary injury is yet to be discovered [4].

The lack of mechanistic insights into neuronal and vascular toxicity in ICH has been an obstacle to understanding ICH treatment. It was known that ICH blood clot lysis would release heme, ferrous ions, ferric ions, and other products that played a significant role in iron-related toxicity. In vitro and in vivo studies had demonstrated that heme and iron ions were associated with perihematomal edema within 24 h and were negative predictors of ICH outcome [5]. Ferrous and ferric ions could initiate lipid peroxidation to create free radicals causing oxidative stress. These free irons also attract macrophages/microglia leading to neuroinflammation and further brain injury [6,7,8]. Thus, targeting iron in ICH management might offer opportunities to improve outcomes.

The idea of iron-chelating agents—commonly used in thalassemic patients—having potency against iron overload and toxicity in ICH has been well-thought-out. These agents bind free tissue iron and increase its elimination through urine [9]. They are able to penetrate the blood–brain barrier and are readily available in brain tissue in a significant amount after intravenous administration. Animal studies proved that deferoxamine helped to reduce hemoglobin-induced neurotoxicity by binding to iron ions and creating a stable, non-toxic complex, thus reducing reactive oxygen species. These successful in vitro and in vivo findings enticed many scientists to advance the study in humans [6,8,10]. 

Some clinical trials showed positive effects of iron-chelation in ICH by evaluating hematoma and edema volume. However, the results are mixed and non-translatable to clinical improvement [11,12]. Clinical improvement in stroke is routinely evaluated with the National Institute of Health Stroke Scale (NIHSS) and modified Rankin Scale (mRS), which could reflect patients’ morbidity. To answer our clinical question, we are intrigued to write this evidence-based case report (EBCR) in the hopes of providing more ICH treatment options in the future.

## 2. Clinical Scenario and Question

Mrs. MF, 42 years old, presented to the emergency room (ER) of Cipto Mangunkusumo Hospital with a sudden right-sided weakness for four hours. A slight slur of speech was initially noticed before the weakness. Eight hours later, she started to struggle with writing and walking. She also complained of a one-week period of a recurring, intermittent pulsating left-sided headache that usually lasted for about 30 min and was responsive to paracetamol. Before being brought to the ER, the patient could understand sentences and respond to commands appropriately. Any occurrence of seizure, numbness, tingling, nausea, or vomiting was denied. The patient had a history of 10-year hypertension with non-compliant treatment consisting of 5 mg amlodipine q.d and 5 mg ramipril q.d. She was a housewife with no smoking and alcohol consumption history. Familial history of stroke, hypertension, and other cardiovascular disease was unknown. 

Vital signs were stable except for high blood pressure of 210/110 mmHg and Glasgow Coma Scale (GCS) score of 14 (Eye 3 Verbal 6 Motor 4). Dysarthria, a central type of right facial nerve palsy, and upper-motor neuron right hemiparesis were found. Other remarkable examination findings included absent right patellar reflex and urinary incontinence. Neither meningeal signs nor sensory deficits were found. Upon arrival, she could answer questions with some difficulties, e.g., questions that needed to be repeated several times. She tended to sleep after answering some questions. Non-contrast head CT scan revealed left temporal intracerebral hemorrhage with an approximate volume of 25 mL. The laboratory examination result was only significant for leukocytosis (11,790 cells/mL). 

She was later diagnosed with acute spontaneous intracerebral hemorrhage. Intravenous mannitol 125 mL q.i.d, intravenous paracetamol 1000 mg t.i.d, oral folic acid 5 mg b.i.d, and oral pyridoxine 10 mg b.i.d. were administered, while an intravenous drip of nicardipine starting from 5 mg/hour was also given. The medical team was concerned about the possibility of rapid hematoma expansion and clinical deterioration in this patient, so they started considering alternative treatments. One of the members suggested trying supplementary iron-chelating agents to prevent hematoma expansions, thus ameliorating the neurological outcome.

**Clinical question:** Does the addition of iron-chelation therapy give better neurological outcomes, measured by mRS or NIHSS within 90 days, for patients diagnosed with ICH than those not receiving additional iron-chelation therapy?

Literature searching was performed on 5 October 2022, on four journal databases (Pubmed, Embase, Cochrane, and Google Scholar) using related terminologies (Appendix A). Our inclusion criteria were (1) meta-analysis and/or systematic review of randomized controlled trials (RCTs) and/or RCT with no restrictions of language and publication time; (2) iron chelating agent of all dosage as clinical intervention, defined as any drug aimed to prevent accumulation of excess body iron including deferoxamine (desferrioxamine), deferiprone, or deferasirox; (3) adult patients (>18 years old) presenting with intracranial hemorrhage, especially intracerebral hemorrhage, but not limited to, subarachnoid hemorrhage, subdural hemorrhage, and intraventricular hemorrhage; and (4) clinical improvement within 3 months after onset measured with National Institutes of Health Stroke Scale (NIHSS) and/or modified Rankin Scale (mRS) scores.

On the other hand, our exclusion criteria consisted of the following: (1) animal or laboratory studies; (2) case reports, case series, opinions, and conference abstracts; (3) trial protocols or ongoing trials; (4) early hematoma evacuation before clinical intervention; (5) studies with insufficient data; (6) original studies included in the eligible systematic reviews or meta-analyses. We double-filtered and screened the search results by title, abstracts, and methods using our inclusion and exclusion criteria. We contacted authors whose full-text papers were not available online. Final eligible research papers were then appraised. Critical appraisal was performed using worksheets for meta-analysis/systematic review and an RCT provided by the Oxford Centre for Evidence-Based Medicine (CEEBM).

## 3. Critical Appraisal

Two meta-analyses, two systematic reviews, and one RCT were included in the final analysis using the previously described search strategy (Figure 1 and Table 1). The critical appraisal of these articles based on validity, importance, and applicability criteria is presented in Table 2 and Table 3. The meta-analysis by Zhao et al. [11] included five RCTs, and most of them were assessed as high-quality. The other meta-analysis from Liu et al. [13] included five RCTs assessed as low- to moderate-quality evidence, and three prospective cohorts considered as high-quality evidence. The systematic review from van der Loo et al. [14] comprised two RCTs, and Zeng et al. [15] included one RCT and cohort in their review. Studies reviewed in each review were assessed as either high or low quality.

All meta-analyses and systematic reviews were clear about their PICOs, which exclusively highlighted the effects of deferoxamine in adult patients with ICH. The clinical questions were also well-reflected in their search strategies, except for Zeng et al. [15] where the strategies were not thoroughly defined leading to non-specific results. Furthermore, Liu et al. [13] limited their search only to fully accessible articles. Summary tables and plots, if any, were adequately provided in all articles. Both meta-analyses mentioned heterogeneity, but only Zhao et al. [11] explained this phenomenon using influence analysis. They also incorporated funnel plot analysis, with no publication bias detected. 

On the other hand, the post hoc analysis of the intracerebral hemorrhage deferoxamine (i-DEF) trial by Wei et al. [16] was also included in our analysis. The article analyzed the efficacy of the 3-day course of intravenous deferoxamine 32 mg/kg per day in ICH patients based on the hematoma volume subgroups (small <10 mL, moderate 10–30 mL, and large >30 mL). By evaluating the i-DEF protocol, randomization and blinding were performed in the study, but there were no clear statements about additional management and placebo between groups. In addition, per protocol approach was used in this post-hoc analysis article with a low drop-out rate (~2.5%). Despite the flaws, we concluded that all the articles filtered were valid with moderate- to high-quality levels. 

Zhao et al. [11] found the optimal dose of desferrioxamine that could improve NIHSS (mean difference 0.24 [95%CI 0.03 to 0.45]) within 90 days after stroke onset, which is 32 mg/kg/day. They also found NIHSS improvement (mean difference 0.26 [95%CI 0.04 to 0.47]) in terms of desferrioxamine administration (intravenous infusion). Because these size effects were not crossing the line of no difference and were homogenous, we deemed this meta-analysis to be important.

The RCT of Wei et al. [16] was also deemed to be important based on our appraisal. Among the ICH patient subgroups based on the hematoma volume, the moderate subgroup was the only subgroup with a positive relative risk reduction (RRR) of 0.287 (Table 3). Meanwhile, its absolute risk reduction (ARR) actually crossed the line of no difference but with a very close lower bound to zero; ARR: 0.166 (95%CI −0.008 to 0.340). The NNT was six, which made it low enough to exert benefit in the subjects. 

We considered the article by Liu et al. [13] as not important owing to the non-significant differences of NIHSS and mRS between deferoxamine and the control group. The systematic reviews of Van der Loo et al. [14] and Zeng et al. [15] were also evaluated as non-clinically important since there were too few trials, which made it difficult to conduct further cumulative quantitative analysis. Finally, we concluded that studies by Zhao et al. [11] and Wei et al. [16] were applicable because the patient’s conditions were relatively similar to study subjects’ characteristics.

## 4. Discussion

Currently, ICH studies focus on the potential benefits of protection from secondary brain injury from reactive oxidative species (ROS), inflammation, and toxicity of erythrocyte lysates. Secondary brain injury in ICH occurs because of the limited capacity of erythrophagocytosis in scavenging blood products, especially when the hematoma volume is progressively increasing. This leads to the accumulation of ferrous iron as the final product of heme breakdown, and later to iron toxicity. A labile iron pool would trigger ferroptosis, a process marked by cell death facilitated with glutathione depletion due to unregulated ferrous iron reacting with superoxide, generating ROS [7,17]. 

Further, hemoglobin and its lysate trigger perihematomal edema as they bind with TLR4 receptors on macrophage and microglia, leading to an inflammatory cascade. Several animal studies showed that substrate injections to mice would cause blood–brain barrier (BBB) breakdown, increased sodium, and increased matrix metalloproteinase-9 (MMP-9)—all components contributing to cerebral edema. These findings imply that intracerebral iron overload may lead ICH patients to poorer prognosis. Liu et al. [18] found that intracerebral iron accumulation was correlated with post-ICH perihematomal edema and brain atrophy. Furthermore, there was a significant positive correlation between iron overload indexed as high ferritin with high mRS score [5,19,20]. Given this mechanistic insight, several cerebral protection alternatives to stop negative effects from erythrocyte lysate have been proposed (Figure 2).

Deferoxamine is an iron-chelating agent approved for treating acute and chronic iron overload. It is a hexadentate molecule with the ability to bind free iron and labile iron pool with ratio of 1:1 [9,10,21]. Deferoxamine is also able to rapidly penetrate the blood–brain barrier and accumulate in brain tissue after systemic administration [6,22]. These properties had brought numerous investigations of its efficacy in ICH animal model, mostly with positive results of reducing iron-mediated oxidative damage, hematoma volume, perihematomal edema, and thus neurological dysfunction. In addition to the direct iron-binding effect, post-ICH secondary injury attenuation by deferoxamine was also suggested to occur via its effects on HO-1 expression and Fenton and Haber–Weiss reactions—important reactions in reactive hydroxyl radical generations [6,23].

Based on theoretical frameworks and findings from preclinical studies, clinical studies were performed to see if benefits from deferoxamine held true in ICH patients. Several trials have shown a promising benefit of deferoxamine in reducing hematoma volume and perihematomal edema with the most obvious decrease observed within 7 days since ICH onset [11]. However, these important findings were not well-translated into neurological outcome as shown by result heterogeneity across studies filtered from our literature searching. We considered studies by Zhao et al. [11] and Wei et al. [16] as valid, important, and reliable because they provided enough evidence to use deferoxamine in ICH patients. Nevertheless, some questions need to be addressed in its administration: deferoxamine dosage, prompt hematoma volume, and measures on neurological outcome.

The meta-analysis by Zhao et al. [11] provided moderate evidence that 32 mg/kg/day deferoxamine given for three consecutive days reduced NIHSS by 0.26 (95% CI 0.04 to 0.47). This dose was in line with the usual dosage utilized in usual indications (25–50 mg/kg/day) [9]. A phase I study of the intracerebral hemorrhage deferoxamine trial (iDEF) reported using deferoxamine with a dosage ranging from 7 to 62 mg/kg/day with a maximum dose of 6000 mg/day [24]. Based on this study, the high-dose deferoxamine in intracerebral hemorrhage (HI-DEF) trial was carried out to seek high-dose deferoxamine (62 kg/mg/day) efficacy in ICH patients [25]. This trial was terminated early due to an increased incidence of acute respiratory distress syndrome (ARDS) [26]. 

The volume of which ICH might be beneficial after deferoxamine therapy should be clearly defined. The previous study included ICH patients with a wide range of hematoma volumes (2.89 to 62.49 mL), which might explain the unsatisfying results it yielded [16,27]. In the importance analysis of the study by Wei et al. [16], we found the moderate volume subgroup (10–30 mL) to have positive RRR and low NNT, but the opposite in small volume (<10 mL) and large volume (>30 mL). Small hemorrhage could have a better resolution and clearance process, with free iron being too minimal to be taken up by iron chelators [12]. In addition, this subgroup had lower baseline severity (NIHSS or mRS); hence, administering iron chelation therapy may cause more harm than benefit [16]. On the other hand, the effect of iron-chelation therapy in large-volume ICH might be hindered by a mass effect that will increase intracranial pressure [12]. With this rationale, we encourage future studies to focus on deferoxamine’s effect on ICH with a hematoma volume of 10–30 mL. 

The last aspect to deal with is the measuring tool of neurological outcome. There was considerable heterogeneity across studies found, regarding how studies defined their outcome. Most of these studies used NIHSS and mRS scores, with varying checkpoints (day 7, day 14, day 90, and day 180). Only one study in the meta-analysis reported significant NIHSS improvement on day 90 post administration of deferoxamine [11]. This effect was not seen in other studies using NIHSS within a shorter period. Meanwhile, there was almost no significant effect on mRS reported across time in other studies. An intriguing report of a significantly higher proportion of good outcomes, defined as mRS 0–2 on day 180 but not on day 90, was demonstrated when ICH patients were stratified by hematoma volume [16]. Our important analysis of the same study indeed showed a positive ARR of good outcome on day 90, but its 95% CI slightly crossed the line of no difference. 

Some possibilities could explain these inconsistent findings. First, a bigger sample size was needed, given that the 95% CI of ARR just slightly crossed the line (−0.008). Another explanation is also the need to use longer observation period. However, this seems unlikely to be true, proven with consistent reports on perihematomal edema growth rate within 72 h as a predictor of mRS ≥3 within 90 days [28]. Finally, the measurement choice in determining neurological outcome might as well affect the results. Despite mRS being chosen in many ICH trials due to being brief, highly inter-rater reliable, and consistent, its grading is variable and ill-defined [27,29]. Wilson et al. [30] recommended using a structured interview of mRS to achieve good inter-rater reliability and low bias. Another common measure, the Barthel Index, might be more responsive and sensitive to outcome changes in patients with more severe presentations [31,32]. The Glasgow Coma Scale is also relevant when used in patients with decreased consciousness [4] but consists of less specific neurological deficits compared to NIHSS. 

There were some adverse effects noted by two previous studies by Selim et al. [27] after deferoxamine administration in their trials, specifically in 32 mg/kg/day dosage, discovered within 90 days after onset, such as anemia, erythema around the administration site, injection extravasation, hypotension, headache, and delirium tremens. The adverse effect occurred in all patients (n = 3) enrolled to receive a 32 mg/kg/day dosage. Judging from the time window of symptom emergence and there being no significant difference in occurrence between two groups within the first 7 days post-administration, these adverse effects were deemed not to be caused by deferoxamine. Taking this premise into account, deferoxamine is an overall safe and well-tolerated iron-chelating agent, with side effects of nausea, hypotension, and abdominal pain in acute use, while visual and auditory toxicity was associated with chronic use and reversibility [9,21,24,27,33].

Considering these factors, we would offer this modality as an additional treatment to the standard emergency care of ICH stroke. The hypothetical algorithm in which deferoxamine could be given in ICH patients was presented on Figure 3. With body weight of 50 kg and a chosen dosage of 32 mg/kg/day, our patient will receive deferoxamine with a total dose of 4800 mg for three days. In Indonesia, deferoxamine is available under the brand Desferal^®^; the total cost needed for three days would be around Rp 1,128,000 (USD 72.9). Since it is not a standardized treatment in hemorrhagic stroke, we are unsure that the National Health Insurance would have deferoxamine covered for our patient’s treatment; thus, the private cost should be provided. Our patient fulfilled the timing criteria for deferoxamine administration; hence, we believe this patient would receive similar benefits from the treatment. The measured hematoma volume also fit the hypothesized range of hematoma volume that could benefit from deferoxamine administration. 

Our recommendation, however, may not be fit for a posterior fossa hemorrhage. The studies we appraised were focused on intracerebral hemorrhage in general, regardless of its site. Posterior fossa hemorrhage owns a tailored recommendation in the AHA/ASA guideline [4], that is surgical intervention in patients with a hemorrhage volume of >15 mL, or conservative therapy in a smaller hemorrhage volume. These recommendations were due to the fact that a confined hemorrhage in the small posterior fossa space may cause rapid deterioration and need more aggressive treatment. Some former studies excluded patients with infratentorial hemorrhage due to these reasons [25,27]. Therefore, we cannot impose our recommendation on these cases.

## 5. Conclusions

Despite the heterogeneity noted across previous studies, iron-chelation therapy, especially with deferoxamine, merits being considered as a therapy modality in patients presenting with ICH. In this clinical scenario, the recommendation of intravenous deferoxamine with a dosage of 32 mg/kg/day for three days could be offered to the patient, as the studies’ demographic and hematoma volume requirements fit our patient. As this therapy is still novel, thorough information about the drug should be carefully conveyed to this patient, especially regarding acute and chronic adverse effects. We also encourage future, more extensive studies on deferoxamine efficacy in ICH patients, focusing on those with moderate hematoma volume (10–30 mL), and with utilization of the Barthel Index and GCS as additional measures of neurological outcome. Some ongoing trials reflect considerable interest in iron-chelating therapy as an ICH therapy modality; the results of which would add insights in regard to its efficacy.

## Figures and Tables

**Figure 1 medicina-59-00453-f001:**
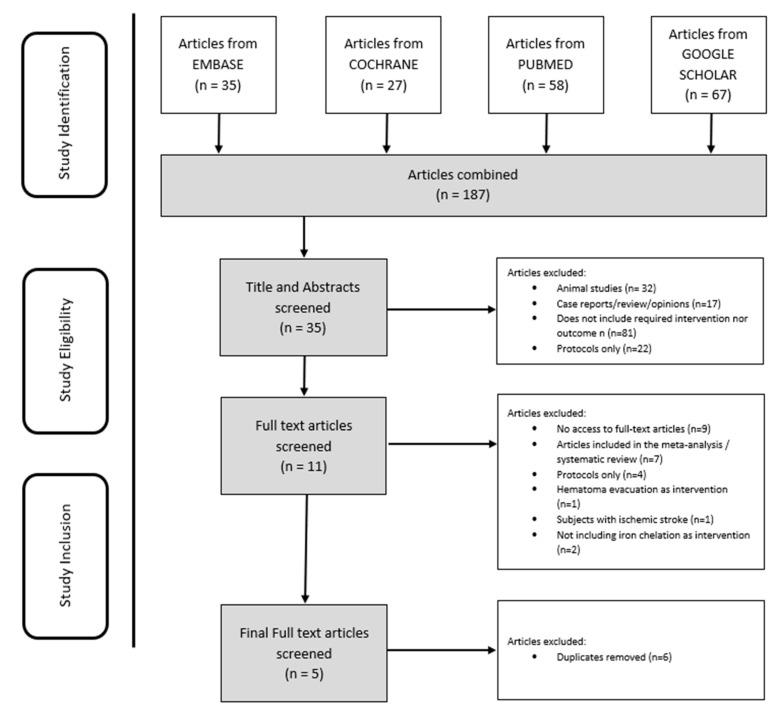
PRISMA Flowchart of Study Selection based on our eligibility criteria.

**Figure 2 medicina-59-00453-f002:**
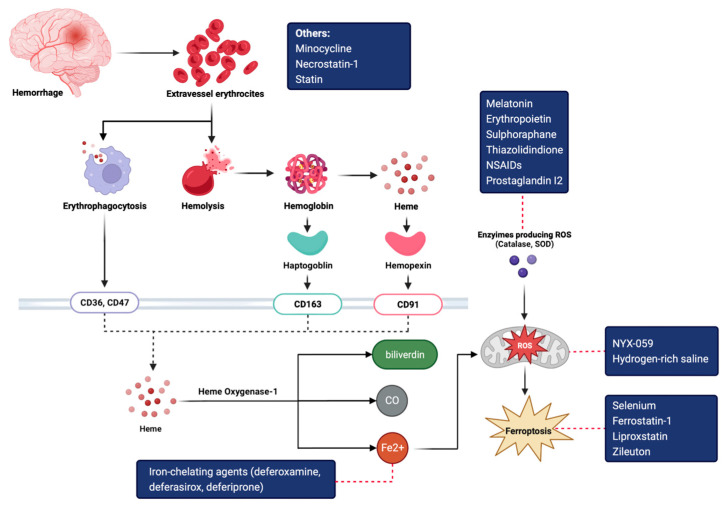
Schematic diagram of iron toxicity mechanism in intracranial hemorrhage with targeted treatment approach (blue box, red dashed line) [7,17]. CO, carbon monoxide; Fe^2+,^ ferrous iron; NSAIDs; non-steroidal anti-inflammatory drugs; ROS, reactive oxidative species; SOD, superoxide dismutase.

**Figure 3 medicina-59-00453-f003:**
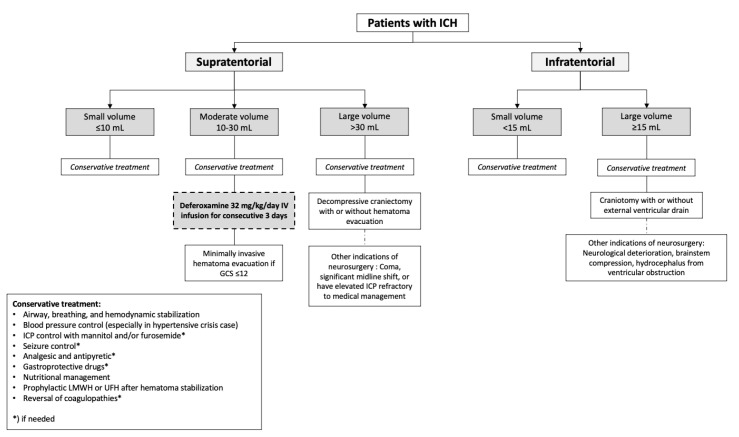
Hypothetical algorithm of ICH treatment with attribution of deferoxamine (dashed box) considering hematoma location and volume. GCS, Glasgow coma scale; ICP, intracranial pressure; LMWH, low-molecular-weight heparin; UFH, unfractionated heparin.

**Table 1 medicina-59-00453-t001:** Characteristics of included studies.

Author	Study Design	N	Population	Intervention	Comparison	Outcome	Level of Evidence
Zhao et al. (2022) [11]	Meta analysis	239	ICH patients	Desferrioxamine	Placebo	Reduction in hemorrhage volume, reduction in perihematomal edema, improvement of neurological function	Ia
Liu et al. (2021) [13]	Meta analysis	619	Patients aged older than 18 years with spontaneous ICH who did not need surgery treatment	Deferoxamine 20–62 mg/kg/day within 24 h after the onset for 3–5 consecutive days	Placebo	Hematoma and edema absorption (primary), neurologic outcome improvement (secondary)	Ia
Van der Loo et al. (2020) [14]	Systematic review	333	Adults with acute stroke	Deferoxamine	Placebo	● Death from all causes at the end of scheduled follow-up ● Good neurological outcome (mRS 0–2) ● Serious adverse events ● Any deaths within the treatment period ● Neurologic impairment scale at baseline and the end of follow-up (NIHSS) ● Relative edema volume ● Quality of life	Ia
Zeng et al. (2018) [15]	Systematic review	71	Patient aged older than 18 years with spontaneous ICH within 24 h confirmed by CT	Intravenous injections of deferoxamine 32 mg/kg/day within 24 h after the onset for 3 consecutive days	Placebo	Hematoma and edema absorption (primary), neurologic outcome improvement and adverse effect (secondary)	Ia
Wei et al. (2022) [16]	RCT	291	Patients aged 18 to 80 years with primary, spontaneous, supratentorial ICH	Deferoxamine 32 mg/kg/day days within 24 h after onset for 3 consecutive	Placebo	mRS assessed on day 90 and 180	Ia

**Table 2 medicina-59-00453-t002:** Critical appraisal for validity and applicability of studies included.

	Zhao et al. (2022) [11]	Liu et al. (2021) [13]	Van der Loo et al. (2021) [14]	Zeng et al. (2018) [15]	Wei et al. (2022) [16]
**Critical appraisal for validity of systematic review/meta analysis**					
Focused questions (PICO)	●	●	●	●	
Using PICO for article searching and selection	●	●	●	●	
All relevant articles are found	●	●	●	●	
Critical appraisal	●	●	●	●	
Inclusion of only high-quality studies	●	●	●	●	
Summary tables and plots	●	●	●	●	
Heterogeneity assessment and explanation	●	●	●	●	
**Critical appraisal for validity of RCT**					
Randomization					●
Similarity at the beginning of the study					●
Equal treatment					●
Intention-to-treat					●
Objective measurement and blinding					●
**Critical appraisal for applicability of studies**					
Clear assessment of patient’s values and preferences	●	N/A	N/A	N/A	●
Suitability for the patient	●				●
Qualitative efficacy differences in some subgroups	●				●

All checklists from Oxford CEEBM. Abbreviations: N/A: Not analyzed; RCT: Randomized controlled trial. Notes: Yes, clearly stated in the study: ● Unclear/variable results: ● No, explicitly stated otherwise: ●.

**Table 3 medicina-59-00453-t003:** Critical appraisal for importance of studies included.

Critical Appraisal for Importance of Systematic Review/Meta Analysis Studies								
Study			Effect Size					Conclusion
Zhao et al. (2022) [11]			Cumulative analysis on NIHSS improvement 90 days after administration: SMD 0.25 (0.05 to 0.45), I^2^ = 0.0%, *p* = 0.992					Important
Liu et al. (2021) [13]			NIHSS evaluated 2 weeks after administration: SMD −3.41 (−8.00 to 1.18), I^2^ = 93%, *p* < 0.00001. Score of mRS less than 3, three months after administration: OR 0.94 (0.61 to 1.43), I^2^ = 0%, *p* = 0.581					Not important
Van der Loo et al. (2020) [14]			Little to no difference of NIHSS evolution in 90 days (placebo vs. deferoxamine: 13 to 4 vs. 13 to 3; *p* = 0.37). Slight reduction in relative perihematomal edema at 15 days (placebo vs. deferoxamine: 1.91 vs. 10.26; *p* = 0.042)					Not important
Zeng et al. (2018) [15]			One RCT reported non-significant results between two groups at 13 and 15 days. One cohort reported greater reduction in deferoxamine group at 7 (mean score 11.7 ± 4.1 vs. 15.1 ± 4.9, *p* < 0.05) and 14 days (mean score 7.4 ± 2.6 vs. 11.8 ± 5.6, *p* < 0.05)					Not important
**Critical Appraisal for Importance of RCT Study**								
**Subgroup**	**CER**	**EER**	**RRR**	**RR**	**ARR**	**NNT**	**CI95%**	**Conclusion**
Overall	0.667	0.639	0.041	0.958	0.028	35.7	−0.830 to 0.139	Not important
Small volume subgroup (<10 mL)	0.500	0.603	−0.206	1.206	0.103	9.7	−0.075 to 0.281	Not important
Moderate volume subgroup (10–30 mL)	0.759	0.593	0.287	0.781	0.166	6.0	−0.008 to 0.340	Important
Large volume subgroup (>30 mL)	0.806	0.960	−0.191	1.191	0.154	6.5	−0.005 to 0.319	Not important

Abbreviations: NIHSS National Institute of Health Stroke Scale; mRS modified Rankin Scale; RCT randomized controlled trial; SMD standardized mean difference; OR odds ratio; CER control event rate; EER experimental event rate; RRR relative risk reduction; RR relative risk; ARR absolute risk reduction; NNT number needed to treat; CI95% 95% confidence interval.

## Data Availability

All data generated as part of this study are included in the article.

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
