# Peer review of "The Role of Iron-Chelating Therapy in Improving Neurological Outcome in Patients with Intracerebral Hemorrhage: Evidence-Based Case Report"

_medicina, 2023, doi:10.3390/medicina59030453_

Round 1
Reviewer 1 Report (Previous Reviewer 1)
Ready for publication.
Author Response
Thank you for your feedback
Reviewer 2 Report (New Reviewer)
Interesting and well-written manuscript.
The Editor should decide if this manuscript is inside the "journal's scope."
Author Response
Thank you for your feedback
Reviewer 3 Report (New Reviewer)
This case report is intriguing and well-written. Please check and revise some typos.
Author Response
Thank you for the feedback given. We have already done revisions on the typos and misspellings
Reviewer 4 Report (New Reviewer)
Enormous advancement has been observed in medical practice(1). The patients with ICH require a comprehensive understanding of the pathophysiology, because their outcomes are still poor.
In this study, the authors searched four databases (Pubmed, 15
Cochrane, Embase, dan Google Scholar) to find studies assessing the efficacy of iron-chelating ther- 16
apy in ICH patients.
Validity, importance, and applicability of the included articles were appraised using worksheets from Oxford Centre for Evidence-Based Medicine . It is well known that the validity in research refers to how accurately a study answers the study question or the strength of the study conclusions(2). They found that two out of five eligible
studies are valid, important, and applicable their patients. The critical appraisal based on validity, importance, and applicability criteria is presented in Table 2 and Table 3. I recommend tthem to discess more about validity process of their study.
They also discussed the volume of which ICH might be beneficial after deferoxamine therapy. However, they did not mentioned about the hemorrhage site. It is clear that posterior fossa hemorrhages have increased morbidity and mortality rate than other sites hemorrhage(3).
References
1. Kanat A, Tsianaka E, Gasenzer ER, Drosos E. Some Interesting Points of Competition of X-Ray using during the Greco-Ottoman War in 1897 and Development of Neurosurgical Radiology: A Reminiscence. Turk Neurosurg. 2022;32(5):877–81.
2. Ozdemir B, Kanat A, Durmaz S, Ersegun Batcik O, Gundogdu H. Introducing a new possible predisposing risk factor for odontoid type 2 fractures after cervical trauma; Ponticulus posticus anomaly of C1 vertebra. J Clin Neurosci. 2022 Nov;96:194–8.
3. Yilmaz A, Musluman AM, Kanat A, Cavusoglu H, Terzi Y, Aydin Y. The correlation between hematoma volume and outcome in ruptured posterior fossa arteriovenous malformations indicates the importance of surgical evacuation of hematomas. Turk Neurosurg. 2011;21(2):152–9.
Author Response
Thank you for the feedback given
First of all, we authors agreed that we did not explore the validity aspect of articles found in our article. We have already added more narrative regarding this under Appraisal of Literatures section
We're also in accord that hemorrhagic site could change the fate of ICH patients receiving deferoxamine. We even found this idea was insightful for us. However, we found insufficient literature in respect of this matter. Some studies even excluded those with infratentorial hemorrhage.1,2 We discussed about this topic at the end of Discussion section
1. Yeatts SD, Palesch YY, Moy CS, Selim M. High Dose Deferoxamine in Intracerebral Hemorrhage (Hi-Def) Trial: Rationale, Design, and Methods. Neurocrit Care. 2013 Oct;19(2):257–66. 2. Selim M, Foster LD, Moy CS, Xi G, Hill MD, Morgenstern LB, et al. Deferoxamine mesylate in patients with intracerebral haemorrhage (i-DEF): a multicentre, randomised, placebo-controlled, double-blind phase 2 trial. The Lancet Neurology. 2019 May;18(5):428–38.
Round 2
Reviewer 4 Report (New Reviewer)
In this paper, the authors considered iron-chelating therapy as an additional therapy for ICH patients, given its benefits and adverse ef fects. More specific studies using larger sample size, focusing on moderate-volume ICH, and using standardized neurological outcomes are encouraged.
The paper does not add anew data.
The revision was not well done
Author Response
According to previous suggestions provided by reviewer #4, we added more narrative regarding validity and hemorrhagic site in the revised version of our article. However, on the latest feedback, the reviewer highlighted on the conclusion of our paper and commented that our revision did not add more value.
Firstly, the reviewer suggested more addition on discussions of validity aspects of articles included in our EBCR. Based on guide given by Oxford CEBM, validity evaluation of a study should focus on the method of the study was conducted. Therefore, we added more discussions regarding this matter. For systematic reviews/meta-analyses we sought for clinical question, searching strategy, result presentation (plot and tables), and publication bias. For randomized controlled trials (RCTs) we tried to find any allocation, performance, and attrition bias. All of these were included in “Appraisal of Literature” section.
Furthermore, the reviewer also recommended us to discuss about hemorrhage site as possible contributing factor in deferoxamine therapy success in ICH patients. After thorough auxiliary literature searching, it seemed that there was lack of studies about medical treatment in posterior fossa hemorrhage, as more invasive treatment such as surgery was more preferential than neuroprotection strategy. Some trials about iron-chelating therapy in ICH even excluded those with such condition. Therefore, there was nothing much to discuss but we still explained it on the “Discussion” section.
Hopefully, our explanation could build the same perception between us authors and reviewers. We also welcomed more elaborated suggestions from reviewer #4 that could be helpful to make the upcoming revision process, if any, easier. In the meantime, we are trying to revise our article in the best way possible
This manuscript is a resubmission of an earlier submission. The following is a list of the peer review reports and author responses from that submission.
Round 1
Reviewer 1 Report
The authors have written an interesting review paper demonstrating how iron-chelating therapy improves neurological outcomes in patients with intracerebral hemorrhage, thus given its benefits and side effects should be considered as an additional therapy for these patients
Just a small suggestion: Please be careful with the format of presenting the number of patients in the abstract. Rather write 1533 than 1.533.